# HIV testing uptake and HIV positivity among presumptive tuberculosis patients in Mandalay, Myanmar, 2014-2017

Khine Wut Yee Kyaw[1,2]*, Nang Thu Thu Kyaw[1,2], Myo Su Kyi[3], Sandar Aye[1‡], Anthony D. Harries[2,4‡], Ajay M. V. Kumar[2,5,6‡], Nay Lynn Oo[1‡], Srinath Satyanarayana[2‡], Si Thu Aung[7‡]

1 International Union Against Tuberculosis and Lung Disease (The Union), Mandalay, Myanmar, 2 International Union Against Tuberculosis and Lung Disease (The Union), Paris, France, 3 Department of Public Health, National Tuberculosis Programme, Myanmar, 4 London School of Hygiene and Tropical Medicine, London, England, United Kingdom, 5 International Union Against Tuberculosis and Lung Disease, The Union South-East Asia Office, New Delhi, India, 6 Yenepoya Medical College, Yenepoya (Deemed to be University), Mangaluru, India, 7 Disease Control Division, Department of Public Health, Nay Pyi Taw, Myanmar

☯ These authors contributed equally to this work.
‡ These authors are joint senior authors on this work. SS and STA are joint senior authors.
* dr.khinewutyeekyaw2015@gmail.com

**Data Availability Statement:** The data contains sensitive presumptive TB patients' information that was obtained from Myanmar's National TB Programme after approval from the relevant

## Abstract

### Introduction

The World Health Organization's framework for TB/HIV collaborative activities recommends provider-initiated HIV testing and counselling (PITC) of patients with presumptive TB. In Myanmar, PITC among presumptive TB patients was started at the TB outpatient department (TB OPD) in Mandalay in 2014. In this study, we assessed the uptake of PITC among presumptive TB patients and the number needed to screen to find one additional HIV positive case, stratified by demographic and clinical characteristics.

### Method

This was a cross-sectional study using routinely collected data of presumptive TB patients who registered for PITC services at the TB OPD between August 2014 and December 2017 in Mandalay.

### Result

Among 21,989 presumptive TB patients registered, 9,796 (44.5%) had known HIV status at registration and 2,763 (28.2%) were people already living with HIV (PLHIV). Of the remainder, 85.3% (10,401/12,193) were newly tested for HIV. Patients <55 years old, those registered in 2014, 2015 and 2017, those employed and those having a history of TB contact had higher uptakes of HIV testing. Among 10,401 patients tested for HIV, 213 (2.1%) patients were newly diagnosed with HIV and this included 147 (69.0%) who were not diagnosed as having TB. The overall prevalence of HIV (previously known and newly diagnosed) among presumptive TB patients was 14.8% (2,976/20,119). The number needed to screen to find

authorities and in-country ethics committee. We have permission to share only aggregate (or pooled) analyzed data but not individual patient wise data. Therefore, the data cannot be made available publicly. However, if anyone is interested in accessing the individual patient wise de-identified data, they are requested to contact the National TB Programme and Institutional Review Board of Myanmar. Mailing address 1: National TB Programme, Disease Control Division, Department of Public Health, Ministry of Health and Sports, Myanmar. Postal code 15011. Phone: +95673421201 or Fax at +95673421201. Mailing address 2: Institutional Review Board, Department of Medical Research, No. 5, Ziwaka Road, Dagon Township, Yangon, Myanmar. Postal code 1191, Phone: +951375457 (ext-118), Email: ercdmr2015@gmail.com (Institutional email for Institutional Review Board of Department of Medical Research, Myanmar).

**Funding:** The author(s) received no specific funding for this work.

**Competing interests:** The authors have declared that no competing interests exist.

one additional HIV case was 48: this number was lower (i.e., a higher yield) among patients aged 35–44 years and among those who were divorced or separated.

## Conclusion

Uptake of HIV testing among eligible presumptive TB patients was high with four out of five presumptive TB patients being tested for HIV. This strategy detected many additional HIV-positive persons, and this included those who were not diagnosed with TB. We strongly recommend that this strategy be implemented nationwide in Myanmar.

## Introduction

Many developing countries are still faced with a dual public health burden of tuberculosis (TB) and human immunodeficiency virus (HIV) infection. Globally, in 2018, 37.9 million people were living with HIV and 10 million people developed TB disease [1,2]. About 251,000 people died of HIV-associated TB [2]. Those with co-infection have high overall morbidity and mortality. The major challenge in dealing with co-infection is that nearly half of those living with HIV and associated TB are unaware of their status and remained untreated [1]. In 2017, only 3.8 out of 6.4 million notified TB cases had a documented HIV test [3]. To reduce this dual burden, the World Health Organization (WHO) in its 2012 policy on TB/HIV collaborative activities recommended routine provider-initiated HIV testing and counselling (PITC) to both diagnosed TB patients and persons with presumptive TB in order to promote early identification and treatment of HIV to reduce the HIV burden in TB patients [4].

PITC is the practice where HIV testing is offered routinely by health care providers to all patients attending health care facilities as a standard component of medical care [5]. Compared to voluntary HIV counselling and testing, which is a patient-initiated process, PITC has been shown to increase the uptake of HIV testing [6,7] leading to early HIV diagnosis, treatment, care and support services. PITC among diagnosed TB patients has become a standard practice in most national TB control programs in the last decade. However, PITC in patients with presumptive TB is not widely implemented globally, especially in low HIV burden countries, due to resource implications [8]. In the high HIV-burden countries of sub-Saharan Africa, studies have found that HIV-prevalence rates in presumptive TB patients are as high as those in diagnosed TB patients, with most of the HIV-positive patients being eligible for antiretroviral therapy (ART) and cotrimoxazole preventive therapy [9–17]. In the countries with low HIV prevalence, studies have found similar results and benefits from HIV testing of patients with presumptive TB [18–21]. Despite this, the uptake of HIV testing has been reported to be low in patients with presumptive TB in India, with context specific reasons including non-availability of HIV test kits, limited human resources, the fact that sputum specimens were sometimes sent to microscopy centers for investigation instead of the patients themselves being referred, lack of health care provider awareness, recording gaps and high workload. [19,22]. However, opt-out HIV testing; in which HIV testing is done unless the patient refuses to do so, has been found to have a higher uptake of HIV testing among patients admitted to the emergency departments of hospitals.[23]

In Myanmar, the national tuberculosis programme (NTP) started TB/HIV collaborative activities in 2005 after the publication of the WHO's interim policy on HIV/TB collaborative activities in 2004. In mid-2014, the Union supported the NTP to initiate PITC in presumptive TB patients at the TB outpatient department (TB OPD) in Mandalay according to 2012 WHO

recommendations [4]. To date, there has been no formal assessment of this service. Therefore, in this study, we assessed 1) the uptake of HIV testing among persons with presumptive TB, 2) demographic and clinical factors associated with HIV testing and 3) numbers needed to test to find one additional HIV positive case stratified by demographic and clinical characteristics among presumptive TB patients.

## Materials and methods

### Study design

This was a cross-sectional study using data routinely collected from the PITC service.

### Setting

**General setting.** Myanmar is a country in South-East Asia with an estimated population of 52 million and is one of the 30 high TB/HIV burden countries (having the highest estimated number of incident TB cases among people living with HIV) [24]. There are 15 regions and states in Myanmar. Mandalay is the third most populated region in Myanmar with 6.2 million people, of whom 35% live in the urban area. There are seven districts in Mandalay region. Mandalay District has 1.7 million population and is administratively divided into seven townships [25].

**Project setting.** The PITC service to presumptive TB patients was started at the TB OPD at the Mandalay General Hospital, a tertiary hospital in Mandalay, in collaboration with the NTP and The International Union Against Tuberculosis and Lung Disease (The Union) in August 2014. This service has been expanded to other health facilities where The Union has been implementing its programmes. This service was also offered to all those referred from other public and private hospitals, township health departments, community-based active case finding projects, HIV clinics and some charity-based clinics. All patients visiting TB OPD are registered in the TB OPD patient register, given a unique serial number and are seen by medical staff. According to the NTP guidelines, a person who has signs and symptoms of TB (cough, fever, weight loss, night sweat, hemoptysis, lymph node enlargement) is considered to have presumptive TB. Persons with presumptive TB undergo diagnostic investigations for TB disease such as sputum microscopy and/or Xpert MTB/RIF, chest radiography and other tests for extrapulmonary TB based on suspected extrapulmonary sites.

At the same time during the first visit, all presumptive TB patients are also referred to the PITC service desk where HIV counselling and testing services are offered. Presumptive TB patients may visit TB OPD multiple times before the TB diagnosis is made but they are referred to PITC only once on their first visit. The unique identifier (created from an OPD serial number and year of registration) is recorded in the PITC register. This service is provided by three nurse counsellors who are supervised by a field associate employed by The Union. According to project standard operational procedures, the presumptive TB patients are eligible to undergo HIV testing if the patient tested HIV negative more than two weeks ago, there was no prior HIV test or no evidence of an HIV-test result in hand. Both HIV testing and TB diagnosis investigations are carried out on the same day.

At the PITC desk, after pretest counselling and receiving individual consent, an HIV screening test using Alere Determine™ HIV-1/2 rapid test is performed by trained nurse counsellors. Before July 2016, if the patient tested HIV-positive, the result was confirmed by Clearview® HIV 1/2 STAT-PAK® rapid test. After July 2016, the confirmatory test was changed to a parallel testing approach (Uni-Gold™ Recombigen® HIV rapid test and STAT-PAK® rapid test simultaneously) according to national guidelines [26,27]. Test results are given to the patient on the same day after post-test counselling. Post-test counselling is carried out by

counsellors if the patient is HIV-negative and by a TB focal person if the patient tests HIV-positive. Patients who test HIV-positive are referred to the Integrated HIV Care (IHC) clinic for further HIV treatment and care using TB/HIV cross referral forms.

When the TB diagnosis is confirmed before or after referral to the IHC clinic, all TB diagnosed patients are referred to their resident townships for the continuation of anti-TB treatment because TB OPD provides only 1–2 weeks of anti-TB treatment to diagnosed TB patients. The referred patients are given standard NTP referral forms and the staff from the TB OPD get in touch with the township to inform about the referral. Patients' demographics, testing and referral information are recorded in the PITC service register. The data from the register is regularly entered into EpiData entry software and extracted to Excel.

## Study population

All presumptive TB patients registered at the PITC service of TB OPD at Mandalay general hospital between August 2014 and December 2017 were included in the study.

## Data variables and sources of data

Data on the following variables were extracted to answer the study objectives: unique identification, known HIV status at registration, registration date, age, sex, marital status, education, occupation, known (self-reported) diabetes mellitus status, presence of cough, presence of fever, presence of night sweats, presence of weight loss, presence of lymph node enlargement, Determine HIV-test result, STAT-PAK HIV-test result, Uni-Gold HIV-test result, TB diagnosis and type of TB. Data were extracted from Excel databases.

## Analysis and statistics

The electronic databases for four years (2014–2017) were combined into one Excel database and imported into STATA (version 14.1) for analysis. Any duplicate entries of patients were identified by combining name, age and sex for those who visited TB OPD more than once as new presumptive TB patients (they would therefore have been given different unique serial numbers) and only the first entries were kept for this study. The final HIV-test result for patients newly tested for HIV was based on the results of Determine HIV-test, STAT-PAK and Uni-Gold according to the HIV testing algorithm applied in the different time periods [26,27]. If patients had negative Determine tests, they were classified as HIV negative. If the Determine test was positive but there was no result from the confirmatory tests, the patient was classified as HIV screening positive. For the presumptive TB patients registered between 2014 and 2016, if the Determine test and STAT-PAK were both positive, the patient was considered as confirmed HIV positive. For those who were registered in 2017, if all three tests were positive, the patient was considered as confirmed HIV-positive and if the Determine test was positive and only one of the confirmatory tests were positive, the result was considered as HIV-inconclusive.

The number needed to screen to find an additional person living with HIV was calculated by dividing the number tested by the number of confirmed HIV positive cases. [N/n where N is the number tested and 'n' is the number who were HIV positive out of those who were tested]. A multivariable binomial log regression model was used to determine the characteristics associated with the uptake of HIV testing among patients with unknown HIV status at registration. A Poisson regression model with robust standard error estimates was used if the binomial log model failed to achieve convergence [28]. The association was then presented as prevalence ratios (PR) and 95% confidence intervals (CI). A $p$-value $< 0.05$ was considered statistically significant for all analysis and this value was included in the final multivariable

model. The multicollinearity was assessed by using a variance inflation factor (>10 was considered as a cutoff for assessing collinearity).

### Ethics approval

We obtained permission to conduct the study from the National Tuberculosis and HIV/AIDS Programme, Department of Public Health, Ministry of Health and Sports, Myanmar. We received ethics approval from the Ethics Review Committee of the Department of Medical Research, Ministry of Health and Sports, Myanmar (ERC/DMR/2017/110) and from the Ethics Advisory Group of the International Union Against Tuberculosis and Lung Disease (The Union), Paris, France (EAG number 44/17). The data was anonymized before the analysis. As this study involved the review of routinely collected data, obtaining informed consent from patients was waived by both ethics committees.

### Results

A total of 21,989 patients with presumptive TB were registered at the PITC service between August 2014 and December 2017. Among them, 12,002 (54.6%) were referred from Mandalay General Hospital, 3,733 (17.0%) from General Practitioners, 2,530 (11.5%) were self-referrals and the remainder 3,724 (16.9%) were referred from other health facilities such as HIV clinics, township health departments and other private and public hospitals. The presence of cough was the most common presentation (65.3%) followed by loss of weight (43.8%). The mean age (standard deviation-SD) was 43 (18) years. Of these patients, 9,796 (44.5%) had a known HIV status at registration with 2,763 (28.2%) being people living with HIV. The remaining 12,193 (55.4%) patients were eligible for HIV testing.

Among 12,193 patients, 10,401 (85.3%) were tested for HIV and 213 (2.1%) were newly diagnosed with HIV, as shown in Fig 1. If we include all persons with inconclusive or screened positive results as being HIV-positive, the proportion who were HIV-positive was 2.8% (95% Confidence interval (CI): 2.5–3.1). If we include all persons with inconclusive or screened positive results as being HIV-negative, the proportion who were HIV positive was 2.0% (95% CI:1.8–2.3). The overall prevalence of HIV (previously known and newly diagnosed) among presumptive TB cases who had an HIV result was 14.8% (2,976/20,119) (95% CI; 14.3%-15.3%).

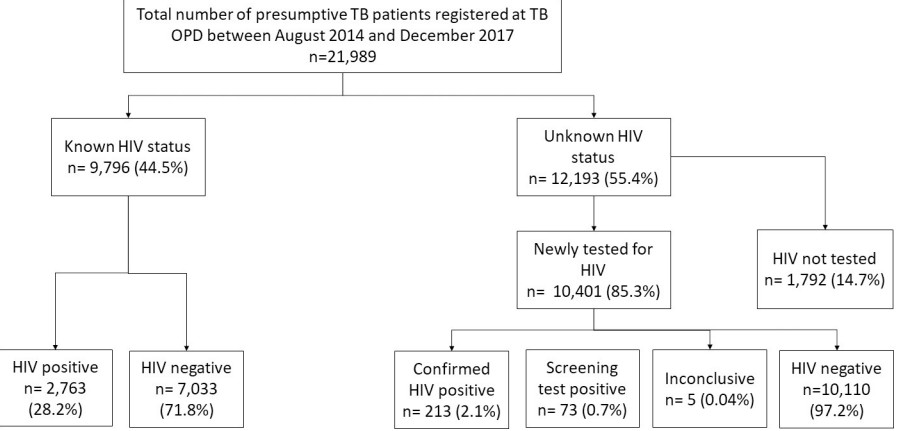

**Fig 1. HIV testing and positivity among presumptive TB patients registered at the PITC services in the TB OPD, Mandalay, August 2014-December 2017.** HIV = Human Immunodeficiency Virus, TB = Tuberculosis, OPD = Outpatient Department, PITC = provider-initiated testing and counseling.

Of the total presumptive TB patients, 6,413 (29.3%) were diagnosed with TB: this included 3,995 (18.2%) pulmonary TB cases, 2,415 (11.0%) extrapulmonary TB cases and 3 (0.01%) MDR-TB cases. Of HIV positive presumptive TB patients (n = 213), only 66 (31.0%) were diagnosed as TB and 147 (69.0%) were not diagnosed with TB.

Table 1 presents the socio-demographic characteristics of presumptive TB patients eligible for HIV testing and their association with the uptake of HIV testing. In adjusted analysis, uptake of HIV testing was significantly higher in patients aged less than 55 years when compared to those aged ≥65 years, those who registered in 2014, 2015 and 2017 when compared to those registered in 2016, those who were employed and those who had a history of TB contact when compared to their counterparts. The number needed to screen to find one additional HIV case was 48 (95% CI: 42–55): this was lower among patients aged 35–44 years and those who were divorced or separated with the numbers needed to test being 20–21 in these categories. The number needed to test was highest among patients aged ≥55 years and patients with known diabetes status as shown in Table 2.

## Discussion

This is the first study in Myanmar to explore the uptake of HIV testing and numbers needed to test to find one additional person living with HIV among presumptive TB patients visiting PITC services at the TB OPD where the majority of presumptive TB patients in Mandalay attended for TB screening. There were five important findings.

First, almost half of the presumptive TB patients knew their HIV status at registration at the TB OPD and 28% of them were already HIV positive. This is because half of the presumptive TB patients were referred from tertiary hospitals where HIV testing was already offered to all admitted patients with common symptoms of HIV and TB before referral to the TB OPD. In addition, medical officers from hospitals and HIV clinics do routine TB screening to all PLHIV and those with symptoms of TB were referred to the TB OPD.

Second, nearly 85% of the patients eligible for HIV testing were tested for HIV. This high uptake was similar to other studies done in Uganda, South India, the Democratic Republic of the Congo and Ethiopia which had testing rates ranging from 85% to 98%, and higher than the studies done in Puducherry District, India (44.6%) and Malawi (56%) [10,18,20,22,29–32]. The possible reason for not being tested for HIV (~15%) in our study is that we offered HIV testing to presumptive TB patients only once during their first visit. Some qualitative studies [conducted among TB patients and contacts of multi-drug resistant TB patients] have reported lack of awareness, stigma, long distances travelled by patients, non-availability of test kits and additional visits to PITC as barriers for HIV testing [33–35].

Third, the patient's age, employment status and history of TB contact were associated with uptake of HIV testing. Younger patients had higher HIV-testing uptake probably due to receiving closer attention from health care providers or self-awareness of HIV transmission compared with older patients. A study done in Africa reported that older people were tested for HIV only when they were symptomatic or if the spouse was HIV positive [36]. Having a job was one of the factors associated with getting tested for HIV. The reason might be because employed people have more interaction with work colleagues which in turn might increase the awareness of HIV and its consequences. In Mandalay, TB contacts are actively traced by community volunteers or Basic Health Staff and are provided with health education for TB screening and HIV testing and this might explain the higher uptake amongst those with TB contacts in our study.

Fourth, through the PITC service, a considerable number of HIV cases were identified among presumptive TB patients with unknown HIV status. This included many patients who

**Table 1. Uptake of HIV testing and associated characteristics among presumptive TB patients with unknown HIV status at registration at PITC services in TB OPD, Mandalay, August 2014-December 2017.**

| Characteristics | Total | (Col%) | HIV testing | (Row %) | PR | (95% CI) | aPR | (95% CI) |
|---|---|---|---|---|---|---|---|---|
| Total | 12,193 | | 10,401 | (85.3) | | | | |
| **Age Group (years)** | | | | | | | | |
| < = 14 | 584 | (4.8) | 512 | (87.7) | 1.16 | (1.11–1.20)* | 1.08 | (1.04–1.12)* |
| 15–34 | 3,866 | (31.7) | 3,529 | (91.3) | 1.20 | (1.17–1.23)* | 1.15 | (1.12–1.18)* |
| 35–44 | 1,778 | (14.6) | 1,590 | (89.4) | 1.18 | (1.15–1.21)* | 1.12 | (1.09–1.15)* |
| 45–54 | 1,838 | (15.1) | 1,590 | (86.5) | 1.14 | (1.11–1.17)* | 1.10 | (1.07–1.13)* |
| 55–64 | 1,800 | (14.8) | 1,415 | (78.6) | 1.04 | (1.00–1.07)* | 1.02 | (0.99–1.05) |
| > = 65 | 2,327 | (19.1) | 1,765 | (75.8) | ref | | ref | |
| **Sex** | | | | | | | | |
| Female | 5,228 | (42.9) | 4437 | (84.9) | ref | | ref | |
| Male | 6,955 | (57.0) | 5961 | (85.7) | 1.01 | (0.99–1.03) | 1 | (0.99–1.02) |
| Not recorded | 10 | (0.1) | 3 | (30.0) | | | | |
| **Marital status** | | | | | | | | |
| Single | 3,852 | (31.6) | 3,438 | (89.3) | 1.15 | (1.11–1.19)* | 1.02 | (0.98–1.06) |
| Married | 7,072 | (58.0) | 5,966 | (84.4) | 1.09 | (1.05–1.13)* | 1.01 | (0.98–1.05) |
| divorced/separated | 137 | (1.1) | 119 | (86.9) | 1.12 | (1.04–1.21)* | 1.01 | (0.94–1.08) |
| Widow | 1,087 | (8.9) | 842 | (77.5) | ref | | ref | |
| Not recorded | 45 | (0.4) | 36 | (80.0) | | | | |
| **Literacy** | | | | | | | | |
| Literate | 11,349 | (93.1) | 9,683 | (85.3) | 1.00 | (0.97–1.03) | | |
| illiterate | 736 | (6.0) | 627 | (85.2) | ref | | | |
| Not recorded | 108 | (0.9) | 91 | (84.3) | | | | |
| **Employment status** | | | | | | | | |
| unemployed | 3,615 | (29.6) | 2,883 | (79.8) | ref | | ref | |
| employed | 8,555 | (70.2) | 7,496 | (87.6) | 1.10 | (1.08–1.12)* | 1.03 | (1.01–1.05)* |
| unknown | 23 | (0.2) | 22 | (95.7) | | | | |
| **TB contact** | | | | | | | | |
| No | 8,730 | (71.6) | 7,211 | (82.6) | ref | | ref | |
| Yes | 3,463 | (28.4) | 3,190 | (92.1) | 1.12 | (1.10–1.13)* | 1.07 | (1.06–1.08)* |
| **Known DM** | | | | | | | | |
| No | 11,577 | (94.9) | 9,890 | (85.6) | 1.03 | (0.99–1.07) | | |
| Yes | 616 | (5.1) | 511 | (83.0) | ref | | | |
| **Year of registration** | | | | | | | | |
| 2014 | 1,850 | (15.2) | 1,719 | (92.9) | 1.33 | (1.30–1.36)* | 1.30 | (1.27–1.34)* |
| 2015 | 4,010 | (32.9) | 3,556 | (88.7) | 1.27 | (1.24–1.30)* | 1.26 | (1.23–1.29)* |
| 2016 | 3,941 | (32.3) | 2,756 | (69.9) | ref | | ref | |
| 2017 | 2,392 | (19.6) | 2,370 | (99.1) | 1.42 | (1.39–1.45)* | 1.38 | (1.35–1.41)* |
| **TB Status** | | | | | | | | |
| Pulmonary TB | 2,245 | (18.4) | 2,112 | (94.1) | 1.14 | (1.12–1.15)* | | |
| Extra Pulmonary TB | 876 | (7.2) | 793 | (90.5) | 1.10 | (1.07–1.12)* | | |
| MDR-TB | 2 | (0.02) | 2 | (100.0) | 1.21 | (1.20–1.22)* | | |
| No TB | 9,070 | (74.4) | 7,494 | (82.6) | ref | | | |

HIV = Human Immunodeficiency Virus, TB = Tuberculosis, PITC = provider-initiated testing and counselling, PR = Prevalence ratio, aPR = adjusted prevalence ratio, OPD = Outpatient Department, CI = confidence interval, Col% = column percentage, Row% = Row percentage, ref = reference, DM = Diabetes Mellitus

* = statistically significant

The variables which were significant in the unadjusted analysis were included in the adjusted model except TB status because it was measured after the outcome (HIV testing).

**Table 2. Characteristics and number needed to screen among presumptive TB patients with unknown HIV status registered at PITC services in TB OPD, Mandalay, August 2014-December 2017.**

| Characteristics | Newly tested for HIV[¥] | New HIV positive cases | Row % | (95% CI) | NNS | 95% CI |
|---|---|---|---|---|---|---|
| Total | 10,323 | 213 | 2.1 | (1.8–2.4) | 48 | (42–55) |
| **Age Group (year)** | | | | | | |
| < = 14 | 511 | 6 | 1.2 | (0.4–2.5) | 85 | (39–231) |
| 15–34 | 3,502 | 80 | 2.3 | (1.8–2.8) | 44 | (35–55) |
| 35–44 | 1,568 | 75 | 4.8 | (3.8–6.0) | 21 | (17–26) |
| 45–54 | 1,574 | 42 | 2.7 | (1.9–3.6) | 37 | (28–52) |
| 55–64 | 1,407 | 9 | 0.6 | (0.3–1.2) | 156 | (83–341) |
| > = 65 | 1,761 | 1 | 0.1 | (0.0–0.3) | 1761 | |
| **Sex** | | | | | | |
| Female | 4,404 | 82 | 1.9 | (1.5–2.3) | 54 | (43–67) |
| Male | 5,916 | 131 | 2.2 | (1.9–2.6) | 45 | (38–54) |
| Not recorded | 3 | 0 | | | | |
| **Marital status** | | | | | | |
| single | 3,426 | 65 | 1.9 | (1.5–2.4) | 53 | (41–68) |
| Married | 5,914 | 124 | 2.1 | (1.7–2.5) | 48 | (40–57) |
| divorced/separated | 118 | 6 | 5.1 | (1.9–10.7) | 20 | (9–53) |
| widow | 830 | 18 | 2.2 | (1.3–3.4) | 46 | (29–78) |
| unknown | 35 | 0 | | | | |
| **Education status** | | | | | | |
| illiterate | 620 | 13 | 2.1 | (1.1–3.6) | 48 | (28–89) |
| Primary | 4,604 | 90 | 2.0 | (1.6–2.4) | 51 | (42–64) |
| secondary | 2,260 | 57 | 2.5 | (1.9–3.3) | 40 | (31–52) |
| high school | 1,293 | 32 | 2.5 | (1.7–3.5) | 40 | (29–59) |
| university/graduated | 1,456 | 18 | 1.2 | (0.7–1.9) | 81 | (51–136) |
| unknown | 90 | 3 | 3.3 | (0.7–9.4) | 30 | (11–144) |
| **Literacy** | | | | | | |
| Literate | 620 | 13 | 2.1 | (1.1–3.6) | 48 | (28–89) |
| illiterate | 9,613 | 197 | 2.0 | (1.8–2.4) | 49 | (43–56) |
| Not recorded | 90 | 3 | 3.3 | (0.7–9.4) | 30 | (11–144) |
| **Employment status** | | | | | | |
| unemployed | 2,865 | 39 | 1.4 | (1.0–1.9) | 73 | (54–103) |
| employed | 7,436 | 174 | 2.3 | (2.0–2.7) | 43 | (37–50) |
| unknown | 22 | 0 | | | | |
| **TB contact** | | | | | | |
| No | 7,147 | 142 | 2.0 | (1.7–2.3) | 50 | (43–60) |
| Yes | 3,176 | 71 | 2.2 | (1.7–2.8) | 45 | (36–57) |
| **Known DM** | | | | | | |
| No | 9,813 | 209 | 2.1 | (1.9–2.4) | 47 | (41–54) |
| Yes | 510 | 4 | 0.8 | (0.2–2.0) | 128 | (50–467) |
| **Year of registration** | | | | | | |
| 2014 | 1,694 | 47 | 2.8 | (2.0–3.7) | 36 | (27–49) |
| 2015 | 3,524 | 91 | 2.6 | (2.1–3.2) | 39 | (32–48) |
| 2016 | 2,740 | 40 | 1.5 | (1.0–2.0) | 69 | (50–96) |
| 2017 | 2,365 | 35 | 1.5 | (1.0–2.1) | 68 | (49–97) |
| **TB Status** | | | | | | |
| Pulmonary TB | 2,095 | 54 | 2.6 | (1.9–3.3) | 39 | (30–51) |

(*Continued*)

**Table 2.** (Continued)

| Characteristics | Newly tested for HIV[¥] | New HIV positive cases | Row % | (95% CI) | NNS | 95% CI |
|---|---|---|---|---|---|---|
| Extra Pulmonary TB | 778 | 12 | 1.5 | (0.8–2.7) | 65 | (37–125) |
| MDR-TB | 2 | - | | | | |
| No TB | 7,448 | 147 | 2.0 | (1.7–2.3) | 51 | (43–60) |

HIV = Human Immunodeficiency Virus, TB = Tuberculosis, PITC = provider-initiated testing and counselling, NNS = number needed to screen, PR = Prevalence ratio, OPD = Outpatient department, DM = Diabetes Mellitus, Row% = Row percentage, CI = confidence interval, MDR = multidrug resistant

[¥] = Those who had completed the HIV testing algorithm (both Determine STAT-PAK between 2014–2016 and Determine, STAT-PAK and Uni-gold tests in 2017)

were not diagnosed with TB. It is clear that these are additional people with HIV who are being identified early as a result of this strategy. Without this strategy, these people will not discover their HIV status unless they have obvious symptoms of HIV or other major opportunistic infections.

Fifth, the number needed to screen to find one additional HIV case was less than those reported by studies done in Puducherry District, India, and more than those reported from Vizianagaram District of India, and this is probably due to the low HIV prevalence among the general population in Myanmar [20,22,37]. In the subgroup analysis, the number needed to screen was lower in the middle age group and in those who were divorced. Emphasizing PITC among these two groups can lead to a higher yield and result in optimal use of resources. Moreover, presumptive TB patients aged over 55 years had a lower yield and they might not be offered provider-initiated HIV testing if there is a need for prioritizing resources.

## Strengths and limitations

The major strength of the study was that we used routinely collected programme data which reflects the ground reality. There were some important limitations. First, there were missing data in some of the variables and we are not sure of the effect of this on our study results. Second, patient factors such as TB contact and DM status were self-reported and therefore there could be some misclassification if there were any deficiencies in recall. Third, we did not know how many patients dropped out between registration for TB diagnosis and the PITC service and the yield we found might have been overestimated. However, we think this gap was minimal because the intervention was implemented in a project setting with dedicated medical officers and other resources.

## Recommendations

As a result of this study, we would like to make a number of recommendations. First, there was a high HIV testing uptake at the PITC service which identified many additional HIV positive cases. Hence, we strongly recommend that this service be expanded nationwide. To have better evidence to support this strategy we would also like to conduct a similar study in another setting. Second, there were 1,770 patients who were not tested for HIV and we recommended that another study is conducted to explore the reasons for this. Third, emphasizing PITC among those who had a higher yield of HIV-positive results (the middle age group) and the withdrawal of PITC among those who had a lower yield (age over 55 years old) should be considered if resources are limited. Fourth, the gaps between registration for TB diagnosis and HIV screening at the PITC service should be analyzed and the reasons for dropping out of the whole cascade, including not being tested for HIV, should be explored. Fifth, it is important to study linkages to HIV care, uptake of ART and CPT, and the outcomes of TB-HIV co-infected

patients and non-TB-HIV patients who are diagnosed from the PITC service. Finally, we recommend future research to focus on understanding the implementation challenges using qualitative research methods.

## Conclusion

This study found a high HIV testing uptake amongst persons presenting with presumptive TB, with nearly four out of five eligible patients undergoing HIV testing. Through this service, many additional people were diagnosed with HIV. Due to the significant benefit of this strategy, we strongly recommend that the strategy is expanded nationwide in Myanmar.

## Supporting information

**S1 Checklist. STROBE checklist.**
(DOCX)

## Acknowledgments

We gratefully acknowledge the support of Department of International Development (DFD), UK, National TB Programme (NTP), TB unit (The Union) and all the presumptive TB patients participated in this study.

## Author Contributions

**Conceptualization:** Khine Wut Yee Kyaw, Nang Thu Thu Kyaw, Myo Su Kyi, Sandar Aye, Anthony D. Harries, Ajay M. V. Kumar, Nay Lynn Oo, Srinath Satyanarayana, Si Thu Aung.

**Data curation:** Khine Wut Yee Kyaw, Nang Thu Thu Kyaw, Myo Su Kyi, Sandar Aye, Nay Lynn Oo, Si Thu Aung.

**Formal analysis:** Khine Wut Yee Kyaw, Srinath Satyanarayana.

**Investigation:** Khine Wut Yee Kyaw, Myo Su Kyi, Anthony D. Harries, Ajay M. V. Kumar, Nay Lynn Oo, Srinath Satyanarayana, Si Thu Aung.

**Methodology:** Khine Wut Yee Kyaw, Nang Thu Thu Kyaw, Myo Su Kyi, Sandar Aye, Anthony D. Harries, Ajay M. V. Kumar, Nay Lynn Oo, Srinath Satyanarayana, Si Thu Aung.

**Project administration:** Khine Wut Yee Kyaw.

**Resources:** Khine Wut Yee Kyaw, Nang Thu Thu Kyaw, Sandar Aye, Anthony D. Harries, Ajay M. V. Kumar, Nay Lynn Oo, Si Thu Aung.

**Software:** Khine Wut Yee Kyaw, Nang Thu Thu Kyaw.

**Supervision:** Khine Wut Yee Kyaw, Nang Thu Thu Kyaw, Anthony D. Harries, Ajay M. V. Kumar, Srinath Satyanarayana, Si Thu Aung.

**Validation:** Khine Wut Yee Kyaw, Nang Thu Thu Kyaw, Myo Su Kyi, Sandar Aye, Ajay M. V. Kumar, Nay Lynn Oo, Srinath Satyanarayana, Si Thu Aung.

**Visualization:** Khine Wut Yee Kyaw, Nang Thu Thu Kyaw, Myo Su Kyi, Sandar Aye, Srinath Satyanarayana.

**Writing – original draft:** Khine Wut Yee Kyaw, Nang Thu Thu Kyaw.

**Writing – review & editing:** Khine Wut Yee Kyaw, Nang Thu Thu Kyaw, Myo Su Kyi, Sandar Aye, Anthony D. Harries, Ajay M. V. Kumar, Nay Lynn Oo, Srinath Satyanarayana, Si Thu Aung.

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
