## [Decision Letter · Decision Letter 0]

22 Jan 2020

PONE-D-19-33429

HIV testing uptake and HIV positivity among presumptive tuberculosis patients in Mandalay, Myanmar, 2014-2017

PLOS ONE

Dear Khine Wut  Yee Kyaw

Thank you for submitting your manuscript to PLOS ONE. After careful consideration, we feel that it has merit but does not fully meet PLOS ONE’s publication criteria as it currently stands. Therefore, we invite you to submit a revised version of the manuscript that addresses the points raised during the review process.

We would appreciate receiving your revised manuscript by 28th February 2020. To enhance the reproducibility of your results, we recommend that if applicable you deposit your laboratory protocols in protocols.io, where a protocol can be assigned its own identifier (DOI) such that it can be cited independently in the future. For instructions see: http://journals.plos.org/plosone/s/submission-guidelines#loc-laboratory-protocols

We look forward to receiving your revised manuscript.

Kind regards,

Kwasi Torpey, MD PhD MPH

Academic Editor

PLOS ONE

2. In ethics statement in the manuscript and in the online submission form, please provide additional information about the patient records/samples used in your retrospective study. Specifically, please ensure that you have discussed whether all data/samples were fully anonymized before you accessed them and/or whether the IRB or ethics committee waived the requirement for informed consent. If patients provided informed written consent to have data/samples from their medical records used in research, please include this information.

4. Thank you for stating the following in the Acknowledgments Section of your manuscript:  "We thank the Department for International Development (DFD), UK for funding the Global Operational Research Fellowship Programme at the International Union Against Tuberculosis and Lung Disease (The Union), Paris, France in which the first author works as an operational research fellow. The funder had no role in study design, analysis and interpretation."

Reviewers' comments:

Reviewer's Responses to Questions

**Comments to the Author**

1. Is the manuscript technically sound, and do the data support the conclusions?

Reviewer #1: Partly

Reviewer #2: Yes

Reviewer #3: Yes

Reviewer #4: Yes

2. Has the statistical analysis been performed appropriately and rigorously? 

Reviewer #1: N/A

Reviewer #2: Yes

Reviewer #3: Yes

Reviewer #4: Yes

3. Have the authors made all data underlying the findings in their manuscript fully available?

Reviewer #1: Yes

Reviewer #2: Yes

Reviewer #3: No

Reviewer #4: No

4. Is the manuscript presented in an intelligible fashion and written in standard English?

Reviewer #1: No

Reviewer #2: No

Reviewer #3: Yes

Reviewer #4: Yes

5. Review Comments to the Author

Reviewer #1: Review statistical errors:

The authors should cross check the figures and summations for alignment; in the abstract, result section and the main text. For, example, total HIV positives from previous pool of presumptive TB cases with known HIV status is 2,763 + 215 = 2,978 (not 2,976). However, 2,976 is used in the main text. The error is likely from use of 213 in the result section against 215 in the abstract.

Grammatical errors are noticed on lines 59, 83, 264…

Areas needing contextual review check:

- WHO does not recognize lymph node enlargement as a clinically high index for presumptive TB (line 114)

- In HTS practices with PITC- counselling is done after the test result - but quick health information is provided to obtain consent (line 125)

- Revise line 134-135. Sentence may be confusing, as the opening paragraph says patients are referred to the PITC service delivery point. Is this in reference to diagnosed co-infected TB/HIV patients? Except if there is a note that both TB and PITC service stations offer TB and HIV services simultaneously - this is the ideal though.

- Line 209: Fig 1 is labelled but located elsewhere.

- The statistical instrument used in table 2 to explain number needed to test to get one positive needs to be better explained and simplified.

Technical content:

- Even though 85.5% of the 10,401 new registrations accepted to do HIV test (positivity: 2.05%), it is not technically sound to conclude on `willingness to test` and `case finding yield` from the whole presumptive TB population of 21,989, when 44.6% (9,818) came with a known HIV status, and as presented in the main text, Mandalay is a referral TB hospital with most of the patients being referred from diverse health facilities, already biased with an `intention-to-test.

- Moreover, being an aggregated data over a couple of years (4), without information on the average duration of the last test to time of registration, combining the newly diagnosed HIV positives and those who came to registration with a known status for impact decision making is technically flawed.

- The recommendation to scale up PITC among presumptive TB case nation-wide could be based on other benefits of TB/HIV collaboration, and not on the yield noticed in this study population, as the data and study setting does not support such conclusion.

Reviewer #2: Overview: “HIV testing uptake and HIV positivity among presumptive tuberculosis patients in Mandalay, Myanmar, 2014-2017” is a cross-sectional study evaluating the outcomes of an HIV provider-initiated testing and counseling service among individuals presenting to a TB outpatient department with presumptive tuberculosis. The authors report that 85.5% of individuals with unknown HIV status agreed to HIV testing with 215 (2%) new infections identified. The majority 69% did not have tuberculosis. They found that the number needed to test to identify one new HIV infection was 48. This study does provide interesting information about a high-risk population in an area with lower rates of HIV testing. Enthusiasm was decreased for this manuscript by multiple grammatical/syntax errors and unclear methodology making it difficult to follow.

Major Comments:

Introduction:

1. The authors discuss Provider-initiated testing and counseling (PITC) but do not mention “Opt-out testing” or opt-out PITC which have been widely adopted. The authors should discuss alternative and more inclusive testing strategies and barriers to their implementation locally.

2. Line 80 – 81, “uptake of HIV testing is low” this is a generalized and vague statement. This should be more specific. Where? Under what circumstances? There certainly areas of high HIV testing uptake and opt-out testing in many TB centers.

Methods:

3. Line 99. The authors state the Myanmar is a “high burden” HIV and TB country. The authors should be more specific and highlight the prevalence and incidence of these.

4. Lines 123 – 126 eligibility for PITC in unclear and should be re-written

5. The definition of “known HIV status” is unclear since this included individuals who were positive and negative. Does this mean previously tested for HIV? If so when?

6. For individuals who previously tested negative when were they offered PITC?

7. How were inconclusive/screened positive HIV tests handled? Is there access to RNA PCR? Were these included in the totals? The authors should consider adding in the screened positive individuals in a sensitivity analysis.

8. How was missing data handled? Were they imputed, deleted, ignored?

Results:

9. Line 196 – it is unclear what definition of “aware of HIV status” means. Why were those that previously tested negative not offered repeat testing?

10. Lines 221 – 222. The authors should state the number needed to treat patients 35-44, those who were divorced/separated, >55, and with DM separately.

Discussion:

11. The authors should discuss how their PITC compare to opt-out testing and opt-out PITC in other settings.

12. The authors should state that self-described HIV negative status is a limitation to this study.

Minor Comments:

1. I would recommend using person-first language throughout. People living with HIV instead of HIV positive.

2. The manuscript should be reviewed for grammatical/syntax errors and multiple run on sentences making it difficult to follow.

3. Line 196. Authors should state the interquartile range instead of the standard deviation.

4. PLHIV should be defined with first use.

Reviewer #3: Review Comments

Manuscript title: HIV testing uptake and HIV positivity among presumptive tuberculosis patients in

Mandalay, Myanmar, 2014-2017:

Overall;

The manuscript is well written and addresses an important subject.

Objectives are clearly stated, and the results are well aligned with the objectives.

The authors describe HTS uptake and HIV positive yield among 21,989 presumptive TB patients served in Mandalay district of Myanmar over a period of three years (2 August 2014 and December). They report 85.5% HIV testing rate among eligible presumptive TB patients and a HIV +ve yield of 2% with TB coinfection rate of 31.6% among newly diagnosed HIV patients. The results reveal 15% non-testing among eligible presumptive TB patients; missed opportunity for identifying 37 HIV +ve clients among the 1,770 eligible presumptive TB patients who were not tested. The significance of this is not adequately discussed yet it is important for improving program performance. Minor inconsistencies are noted between the results section of the abstract vs the manuscript body.

Specific comments:

Line 45: HIV rate reported to 2 decimal places and 1 decimal place (need to be consistent with decimal places) e.g. Line 45-46: Among 10,401 patients tested for HIV, 215 (2.05%) patients were newly

diagnosed with HIV and 147 (69.0%) were among those who were not diagnosed as TB.

Line 59: …the… appearing before 10 million seems to be misplaced

Figure 1: It appears that 22 individuals with unrecorded HIV status were classified as “Known HIV status”…review this

Results

Line 46-47: The overall prevalence of HIV (previously known and newly diagnosed) among presumptive TB

patients was 14.7% (2,976/20,192). Please review the numerator and denominator for this calculation based on the cascade presented in Figure 1. The denominator should only include presumptive TB patients with known HIV status at baseline (9,818) and after PITC (10,401) = 20,219; While the numerator should only include known positives at baseline (2,763) and newly tested positives (215 or 213) = 2,978; PITC clients missing HIV test should be excluded from the denominator.

Line 171-172: The number needed to test to find an additional person living with HIV was calculated by

dividing 100 by the proportion of confirmed HIV positive cases. Apply 215/10,401 to re-affirm the validity of this statement.

Line 197-198: …… and the HIV result not recorded for 22 (0.2%) patients. Consider deleting this statement if the clients with unrecorded HIV status were treated as unknown status. It may confuse readers.

Line 200-201

Among 12,171 patients, 10,401 (85.5%) were tested for HIV and 213 (2.0%) were newly diagnosed as HIV, as shown in Fig 1. The number tested positive is reported as 215 (2.05%) in the abstract (Line 45-46). Please resolve this inconsistency.

Figure 1 presents 22 clients with known but unrecorded HIV status; were these treated as +ve or -ve in the analysis. It may confuse readers. Please review to improve clarity.

Similarly, table 1 Presents 73 patients with unconfirmed HIV +ve positives results and 5 with inconclusive results. It would be helpful to explain how these were ultimately classified for purposes of HIV treatment.

Line 203: Refer to Fig. 1 and re-affirm how its contents contributed to the denominator for estimating the overall prevalence of HIV among presumptive TB patients.

Line 240: Spell out NNS in full under the key for table 2.

Discussion

Line 264-265: ……remaining ~15% patients were not tested 265 for HIV in the study. A possible explanation for this is an important gap stated in line 118-120: The presumptive TB patients may visit TB OPD multiple times before the diagnosis is made but they are referred to PITC only on their first visit. Multiple referrals to PITC in the course of their diagnostic work up for TB may improve HIV testing uptake. This may is a possible programmatic recommendation to address missed opportunities for identification of 37 HIV +ve clients among the 1770 presumptive TB patients who did not undergo PITC.

Line 282-283:

..It is clear that those are additional HIV cases which could be identified early due to implementation of this strategy. The study did not include viral loads or any other measures of time since HIV infection. Therefore, early identification is possible but not clear. Consider editing the sentence to tone this assertion.

Reviewer #4: Review comments

Reviewer: Joseph Mugisha Okello

Title for manuscript: HIV testing uptake and HIV positivity among presumptive tuberculosis patients in Mandalay, Myanmar, 2014-2017

General comment

1. This important manuscript covers an important subject on HIV testing uptake and HIV positivity among presumptive tuberculosis patients. As the authors rightly, say, HIV testing uptake is still low in patients with presumptive TB. Therefore this manuscript generates more literature on this important topic

2. The manuscript is well written, and coherent which makes it easy to read. My only concern is why the authors never followed up the 1770 patients that never tested for HIV. In the discussion, the authors talk about the reasons (from qualitative studies) why patients with TB do not test for HIV. However, since the study setting looks to be organised, it would have been better to follow up the 1770 patients (15%) that never tested for HIV to study the reasons why.

Specific comments

1. Data for this study was obtained from a TB program. Although the authors got ethical clearance from the National TB and HIV/AIDS programme and the ethics review committee, there is no mention in the manuscript on whether the authors received informed consent from the study participants/patients.

2. It is important that the authors give some details on how patients co-infected with TB and HIV are managed or how and where they are referred.

3. In addition, the authors should give details on how the program follows TB patients referred to the resident townships for continuation of anti-TB treatment.

6. PLOS authors have the option to publish the peer review history of their article (what does this mean?). If published, this will include your full peer review and any attached files.

Reviewer #1: No

Reviewer #2: No

Reviewer #3: No

Reviewer #4: Yes: Dr Joseph Mugisha Okello (PhD), MRC/UVRI and LSHTM Uganda Research Unit

---

## [Author Response · Author response to Decision Letter 0]

12 May 2020

Dear Editor,

We thank you for the comments of the reviewers. We have been carefully through the review and have revised our paper accordingly in track change mode. We have attempted to address all the comments of the reviewers and give a point-by-point response. We submit a tracked-changes and a clean version of the paper. The line number stated in response are from tracked changes version. We hope this revised paper may now be acceptable for publication in Plos One.

On behalf of all authors,

Khine Wut Yee Kyaw

Author’s response: Thank you. We have checked the revised manuscript.

2. In ethics statement in the manuscript and in the online submission form, please provide additional information about the patient records/samples used in your retrospective study. Specifically, please ensure that you have discussed whether all data/samples were fully anonymized before you accessed them and/or whether the IRB or ethics committee waived the requirement for informed consent. If patients provided informed written consent to have data/samples from their medical records used in research, please include this information.

Author’s response: Thank you. We have carefully checked and we have revised in the manuscript. [line number 197]

Author’s Response: Thank you. We have included the information on Data availability in the cover letter.

4. Thank you for stating the following in the Acknowledgments Section of your manuscript: "We thank the Department for International Development (DFD), UK for funding the Global Operational Research Fellowship Programme at the International Union Against Tuberculosis and Lung Disease (The Union), Paris, France in which the first author works as an operational research fellow. The funder had no role in study design, analysis and interpretation."

Author’s Response: Thank you. We have revised in the manuscript. [line number 340-349]

Author’s Response: Thank you. We have included captions for supporting information file at the end of the manuscript. [line number 369]

 

Reviewer #1: Review statistical errors:

1. Reviewer comment: The authors should cross check the figures and summations for alignment; in the abstract, result section and the main text. For, example, total HIV positives from previous pool of presumptive TB cases with known HIV status is 2,763 + 215 = 2,978 (not 2,976). However, 2,976 is used in the main text. The error is likely from use of 213 in the result section against 215 in the abstract.

Author’s response: Thank you for this comment. We have revised in the manuscript. [line number 45]

2. Reviewer comment: Grammatical errors are noticed on lines 59, 83, 264…

Author’s response: Thank you. The grammatical errors have been corrected. [line number 59, 81-86, 278-281]

Areas needing contextual review check:

3. Reviewer comment: WHO does not recognize lymph node enlargement as a clinically high index for presumptive TB (line 114)

Author’s response: Thank you for this comment. This was accordingly to the criteria set by National TB Program but it is not available in published guidelines. Therefore, we have removed the reference and revised the sentence. [line number 120]

4. Reviewer comment: In HTS practices with PITC- counselling is done after the test result - but quick health information is provided to obtain consent (line 125)

Author’s response: Thank you. We have included about pre-test counselling in line number 135. 

5. Reviewer comment: Revise line 134-135. Sentence may be confusing, as the opening paragraph says patients are referred to the PITC service delivery point. Is this in reference to diagnosed co-infected TB/HIV patients? Except if there is a note that both TB and PITC service stations offer TB and HIV services simultaneously - this is the ideal though.

Author’s response: Thank you. The PICT service counter is in TB OPD and patients are not referred to other service delivery point. We have changed the word “counter” to “desk” to give clear information. [line number 125, 135]

6. Reviewer comment: Line 209: Fig 1 is labelled but located elsewhere.

Author’s response: The figure was submitted as a separate TIFF file according to the journal guideline and it is available in submission system.

7. Reviewer comment: The statistical instrument used in table 2 to explain number needed to test to get one positive needs to be better explained and simplified.

Author’s response: Thank you. We have revised it as followed. “The number needed to screen to find an additional person living with HIV was calculated by dividing the number tested by the number of confirmed HIV positive cases [N/n where N is the number tested and ‘n’ is the number who were HIV positive of those who were tested]” [line number 180-182]

Technical content:

8. Reviewer comment: Even though 85.5% of the 10,401 new registrations accepted to do HIV test (positivity: 2.05%), it is not technically sound to conclude on `willingness to test` and `case finding yield` from the whole presumptive TB population of 21,989, when 44.6% (9,818) came with a known HIV status, and as presented in the main text, Mandalay is a referral TB hospital with most of the patients being referred from diverse health facilities, already biased with an `intention-to-test.

 Author’s response: Thank you. We agree on this point. We have now clarified in the manuscript that the results on ‘uptake’ and ‘positivity’ are on those presumptive TB patients who were found eligible for HIV testing at the tertiary care hospital and but not among all the presumptive TB patients who were referred or who reached the TB OPD. The data contained in table 2 is only of those presumptive patients who did not know their HIV status at the time of registration at the tertiary care hospital. We agree that the uptake of HIV testing could be higher in presumptive TB patients attending tertiary care setting than when compared to presumptive TB patients found in the community. Therefore, we have included a sentence to reflect this aspect in the discussion section of the manuscript. [line number 324-326]

9. Reviewer comment: Moreover, being an aggregated data over a couple of years (4), without information on the average duration of the last test to time of registration, combining the newly diagnosed HIV positives and those who came to registration with a known status for impact decision making is technically flawed.

Author’s response: Thank you for this comment. We wish to clarify that though this study contains aggregate numbers, we have taken HIV positivity only among those who were found eligible for HIV testing. We have excluded persons whose HIV status was already known. We, therefore, feel that study is not technically flawed as commented by the reviewer. 

10. The recommendation to scale up PITC among presumptive TB case nation-wide could be based on other benefits of TB/HIV collaboration, and not on the yield noticed in this study population, as the data and study setting does not support such conclusion.

Author’s response: Thank you. We agree that the country-wide scale-up of PITC should be based on several considerations and studies (such as ours) provides partial evidence in support of this policy. We have reflected this sentiment in the discussion/conclusion section of the manuscript. [line number 324-326] 

Reviewer #2: 

1. Reviewer comment: Overview: “HIV testing uptake and HIV positivity among presumptive tuberculosis patients in Mandalay, Myanmar, 2014-2017” is a cross-sectional study evaluating the outcomes of an HIV provider-initiated testing and counselling service among individuals presenting to a TB outpatient department with presumptive tuberculosis. The authors report that 85.5% of individuals with unknown HIV status agreed to HIV testing with 215 (2%) new infections identified. The majority 69% did not have tuberculosis. They found that the number needed to test to identify one new HIV infection was 48. This study does provide interesting information about a high-risk population in an area with lower rates of HIV testing. Enthusiasm was decreased for this manuscript by multiple grammatical/syntax errors and unclear methodology making it difficult to follow.

Author’s response: Thank you. We have carefully reviewed the whole manuscript and the grammatical/syntax errors were corrected accordingly.

Major Comments:

Introduction:

2. Reviewer comment: The authors discuss Provider-initiated testing and counselling (PITC) but do not mention “Opt-out testing” or opt-out PITC which have been widely adopted. The authors should discuss alternative and more inclusive testing strategies and barriers to their implementation locally.

Author’s response: Thank you for this comment. We have mentioned about “Opt-out testing” for HIV among patients admitted at emergency department in line number 84 and 85.

3. Reviewer comment: Line 80 – 81, “uptake of HIV testing is low” this is a generalized and vague statement. This should be more specific. Where? Under what circumstances? There certainly areas of high HIV testing uptake and opt-out testing in many TB centers.

Author’s response: Thank you for this comment. We have revised in the manuscript and have clarified about where and under what circumstances the HIV testing is low. [line number 80 to 85] 

4. Reviewer comment: Methods: Line 99. The authors state the Myanmar is a “high burden” HIV and TB country. The authors should be more specific and highlight the prevalence and incidence of these.

Author’s response: Thank you for this comment. We wanted to say “highest TB/HIV burden countries” and this has been revised in the manuscript with the supporting information and the corresponding reference. [line number 103-104]

5. Reviewer comment: Lines 123 – 126 eligibility for PITC in unclear and should be re-written

Author’s response: Thank you. We have revised in the manuscript. [line number 131-133]

6. Reviewer comment: The definition of “known HIV status” is unclear since this included individual who were positive and negative. Does this mean previously tested for HIV? If so when?

Author’s response: Thank you. According to project SOP, the patients are categorized as “Known HIV status” when they had HIV positive result in hand or evidence of receiving HIV care at HIV care facilities or they had tested HIV negative within two weeks with the result in hand. 

7. Reviewer comment: For individuals who previously tested negative when were they offered PITC?

Author’s response: Thank you for this comment. They were not offered HIV testing if they were previously tested HIV negative within 2 weeks. Those who were previously tested negative more than two weeks ago were offered PITC and they were categorized as “Unknown HIV status”. 

8. Reviewer comment: How were inconclusive/screened positive HIV tests handled? Is there access to RNA PCR? Were these included in the totals? The authors should consider adding in the screened positive individuals in a sensitivity analysis.

Author’s response: The patients who had inconclusive or screening positive result should be re-tested after 14 days according to National guideline but unfortunately it was not captured in our data. HIV RNA PCR was not used for diagnosis confirmation in our setting. In this analysis, those who did not complete HIV testing algorithm were excluded in NNS calculation (table 2). We have done the sensitivity analysis and presented in the revised manuscript. If we assume all persons with inconclusive or screened positive results as positive, then the proportion positive is 2.8% (95% CI: 2.5-3.1) and if all persons with inconclusive or screened positive missing results were negative then the proportion HIV positive is 2.0% (95% CI:1.8-2.3). [line number 211-214]

9. How was missing data handled? Were they imputed, deleted, ignored?

Author’s response: In our study the number of persons with missing data (either age, sex, education, occupation, marital status, diabetes status and TB contact) were 381/21,989 (1.7%). As the amount of missing data for each variable was less than 2%, they were excluded in the regression analysis (Table 1). As you can see in the table 2, the presumptive TB patients who had missing data were shown and the NNS was calculated for them. Only those who had invalid HIV results (such as HIV screening test positive but no confirmatory test and those who had inconclusive HIV result) were excluded in NNS calculation. 

10. Reviewer comment: Results: Line 196 – it is unclear what definition of “aware of HIV status” means. Why were those that previously tested negative not offered repeat testing?

Author’s response: Thank you for this comment. We have changed the word to “known HIV status” from the “aware of HIV status” to be consistent and clear. This includes people who had HIV positive result in hand or evidence of receiving HIV care at HIV care facility or those who had HIV negative result which was tested within 2 weeks. Those who have HIV negative result which was tested. 

11. Reviewer comment: Lines 221 – 222. The authors should state the number needed to treat patients 35-44, those who were divorced/separated, >55, and with DM separately.

Author’s response: Thank you. We did say separately for those who had low NNS (35-44 years old and those who were divorced and separated) and those who had high NNS (people aged more than 55 years and those who know DM status).

12. Discussion: Reviewer comment: The authors should discuss how their PITC compare to opt-out testing and opt-out PITC in other settings.

Author’s response: Thank you. We have compared our findings on HIV testing uptake with other studies conducted in Uganda, Democratic Republic of the Congo, Ethiopia, India and Malawi. [line number 276-279].

13. Reviewer comment: The authors should state that self-described HIV negative status is a limitation to this study.

Author’s response: Thank you. We did not include known HIV negative status as a limitation of the study because the status was confirmed by verification with the HIV test result done within last 2 weeks rather than self-described.

14. Reviewer comment: Minor Comments: 1.I would recommend using person-first language throughout. People living with HIV instead of HIV positive.

Author’s response: Thank you. We have carefully checked and revised accordingly.

15. Reviewer comment: The manuscript should be reviewed for grammatical/syntax errors and multiple run on sentences making it difficult to follow.

Author’s response: We have carefully reviewed the grammatical/syntax errors and necessary changes have been made.

16. Reviewer comment: 3. Line 196. Authors should state the interquartile range instead of the standard deviation.

Author’s response: Thank you. As we used “mean” for measurement of central tendency, we want to keep standard deviation for dispersion.

17. Reviewer comment: 4. PLHIV should be defined with first use.

Author’s response: Thank you. We have revised in the manuscript. [line number 42]

Reviewer #3: Review Comments

1. Reviewer comment: Manuscript title: HIV testing uptake and HIV positivity among presumptive tuberculosis patients in Mandalay, Myanmar, 2014-2017: Overall;

The manuscript is well written and addresses an important subject.

Objectives are clearly stated, and the results are well aligned with the objectives.

The authors describe HTS uptake and HIV positive yield among 21,989 presumptive TB patients served in Mandalay district of Myanmar over a period of three years (2 August 2014 and December). They report 85.5% HIV testing rate among eligible presumptive TB patients and a HIV +ve yield of 2% with TB coinfection rate of 31.6% among newly diagnosed HIV patients. The results reveal 15% non-testing among eligible presumptive TB patients; missed opportunity for identifying 37 HIV +ve clients among the 1,770 eligible presumptive TB patients who were not tested. The significance of this is not adequately discussed yet it is important for improving program performance. Minor inconsistencies are noted between the results section of the abstract vs the manuscript body.

Authors’ response: We thank the reviewer for these encouraging comments. We have made several revisions in the discussion section of the manuscript. We hope that with these revisions your concern ‘The significance of this is not adequately discussed’ is addressed in the revised version of the manuscript. [line number 398-300]

Specific comments:

2. Reviewer comment: Line 45: HIV rate reported to 2 decimal places and 1 decimal place (need to be consistent with decimal places) e.g. Line 45-46: Among 10,401 patients tested for HIV, 215 (2.05%) patients were newly diagnosed with HIV and 147 (69.0%) were among those who were not diagnosed as TB.

Authors’ response: Thank you. We have carefully checked and revised in the manuscript. We kept one decimal place for proportion and two decimal places for prevalence ratio on purpose.

3. Reviewer comment: Line 59: …the… appearing before 10 million seems to be misplaced

Response: Thank you. We have revised in the manuscript. [line number 59]

4. Reviewer comment: Figure 1: It appears that 22 individuals with unrecorded HIV status were classified as “Known HIV status” review this. 

Author’s response: Thank you for this comment. We agreed this point and the presumptive TB patients with unrecorded HIV status are now recategorized as “Unknown HIV status”. We have revised the figure 1 and result narrative accordingly. Due to implication of this changes, we have also revised the table 1.

5. Reviewer comment: Results Line 46-47: The overall prevalence of HIV (previously known and newly diagnosed) among presumptive TB patients was 14.7% (2,976/20,192). Please review the numerator and denominator for this calculation based on the cascade presented in Figure 1. The denominator should only include presumptive TB patients with known HIV status at baseline (9,818) and after PITC (10,401) = 20,219; While the numerator should only include known positives at baseline (2,763) and newly tested positives (215 or 213) = 2,978; PITC clients missing HIV test should be excluded from the denominator.

Author’s response: We have revised our calculation for overall prevalence of HIV. Now the denominator is (Known HIV status, 9796 + newly tested for HIV, 10,401) – (HIV result inconclusive, 5 + HIV screening positive only, 73) = 20,119. The numerator is 2,763 + 213 = 2,976. 

6. Reviewer comment: Line 171-172: The number needed to test to find an additional person living with HIV was calculated by dividing 100 by the proportion of confirmed HIV positive cases. Apply 215/10,401 to re-affirm the validity of this statement.

Author’s response: Thank you for this comment. We wish to clarify that 78 persons (73 who were positive in screening test but no confirmatory test and 5 patients with inconclusive results) were excluded from the NNS calculations. As it can be seen in table 2, the denominator for calculating the proportion of new HIV positive was 10,323 because we took only those who completed HIV testing cascade according to different national guideline adopted in different time periods. (¥ = Those who had completed HIV testing algorithm (both Determine STAT-PAK between 2014-2016 and Determine, STAT-PAK and Uni-gold tests in 2017). Therefore, we got 2.1 by dividing 213 by 10,323. The number needed to test was calculated by dividing 100 by 2.1. We revised a sentence to better explain the NNS calculation as dividing the number tested by the number of confirmed HIV positive cases in the revised manuscript. [line number 180-182].

7. Reviewer comment: Line 197-198: …… and the HIV result not recorded for 22 (0.2%) patients. Consider deleting this statement if the clients with unrecorded HIV status were treated as unknown status. It may confuse readers.

Author’s response: Thank you. Patient with unrecorded HIV status were taken as “unknown status” and we have deleted this statement in the revised manuscript. [line number 208]

8. Reviewer comment: Line 200-201 Among 12,171 patients, 10,401 (85.5%) were tested for HIV and 213 (2.0%) were newly diagnosed as HIV, as shown in Fig 1. The number tested positive is reported as 215 (2.05%) in the abstract (Line 45-46). Please resolve this inconsistency.

Author’s response: Thank you. This is our mistake and we have corrected it in this revised manuscript. [line number 45]

9. Reviewer comment: Figure 1 presents 22 clients with known but unrecorded HIV status; were these treated as +ve or -ve in the analysis. It may confuse readers. Please review to improve clarity. Similarly, table 1 Presents 73 patients with unconfirmed HIV +ve positives results and 5 with inconclusive results. It would be helpful to explain how these were ultimately classified for purposes of HIV treatment.

Author’s response: Twenty-two patient with unrecorded HIV status are now categorized as “unknown HIV status” and they were not classified as HIV positive nor negative. The patients who had inconclusive or screening positive result should be re-tested after 14 days according to National guideline but unfortunately it was not captured in our data and those 78 patients were not included in our analysis.

10. Reviewer comment: Line 203: Refer to Fig. 1 and re-affirm how its contents contributed to the denominator for estimating the overall prevalence of HIV among presumptive TB patients.

Author’s response: denominator is (Known HIV status, 9796 + newly tested for HIV, 10,401) – (HIV result inconclusive, 5 + HIV screening positive only, 73) = 20,119. The numerator is 2,763 + 213 = 2,976. 

11. Reviewer comment: Line 240: Spell out NNS in full under the key for table 2.

Author’s response: Thank you for this comment. We have revised in the manuscript. [line number 256]

12. Reviewer comment: Discussion Line 264-265: ……remaining ~15% patients were not tested for HIV in the study. A possible explanation for this is an important gap stated in line 118-120: The presumptive TB patients may visit TB OPD multiple times before the diagnosis is made but they are referred to PITC only on their first visit. Multiple referrals to PITC in the course of their diagnostic work up for TB may improve HIV testing uptake. This may is a possible programmatic recommendation to address missed opportunities for identification of 37 HIV +ve clients among the 1770 presumptive TB patients who did not undergo PITC.

Author’s response: Thank you for this insightful comment. We have added this in the discussion of the manuscript. [line number 275-278]

13. Reviewer comment: Line 282-283: It is clear that those are additional HIV cases which could be identified early due to implementation of this strategy. The study did not include viral loads or any other measures of time since HIV infection. Therefore, early identification is possible but not clear. Consider editing the sentence to tone this assertion.

Author’s response: Thank you. Although we did not viral load and CD4 data to determine the stage of HIV infection, we still believe that 147 presumptive TB patients without TB were diagnosed HIV early due to this strategy. Otherwise they will not know their HIV status unless they have obvious symptoms of HIV or other major opportunistic infections. We have explained this in the revised manuscript. [line number 297-299] 

Reviewer #4: Review comments

Reviewer: Joseph Mugisha Okello

Title for manuscript: HIV testing uptake and HIV positivity among presumptive tuberculosis patients in Mandalay, Myanmar, 2014-2017

General comment

1. Reviewer comment: This important manuscript covers an important subject on HIV testing uptake and HIV positivity among presumptive tuberculosis patients. As the authors rightly, say, HIV testing uptake is still low in patients with presumptive TB. Therefore this manuscript generates more literature on this important topic

Author’s response: Thank you for the encouraging comment.

2. Reviewer comment: The manuscript is well written, and coherent which makes it easy to read. My only concern is why the authors never followed up the 1770 patients that never tested for HIV. In the discussion, the authors talk about the reasons (from qualitative studies) why patients with TB do not test for HIV. However, since the study setting looks to be organised, it would have been better to follow up the 1770 patients (15%) that never tested for HIV to study the reasons why.

Author’s response: Thank you for very practical comments. As it is beyond our study objectives, we were not able to follow up these 1,770 patients. This is an area for future research. 

Specific comments

3. Reviewer comment: Data for this study was obtained from a TB program. Although the authors got ethical clearance from the National TB and HIV/AIDS programme and the ethics review committee, there is no mention in the manuscript on whether the authors received informed consent from the study participants/patients.

Author’s response: Thank you. We have included a statement for this in the manuscript. [line number 196-198]

4. Reviewer comment: It is important that the authors give some details on how patients co-infected with TB and HIV are managed or how and where they are referred.

Author’s response: Thank you. The details on how patients are referred to their respective treatment centres are described in project setting. As we don’t have specific health facilities for receiving co-infected patients, we referred them to separate HIV and TB care facilities like we do for other patients with either of the two infections. [line number 141-149]

5. Reviewer comment: In addition, the authors should give details on how the program follows TB patients referred to the resident townships for continuation of anti-TB treatment.

Author’s response: Thank you. In the routine setting, the TB patients were referred to the resident townships by using NTP’s referral form. Since the facility where the study conducted was not actively checked for continuation of anti-TB treatment. Therefore, we did not include this in the manuscript.

---

## [Editor Report · Decision Letter 1]

19 May 2020

PONE-D-19-33429R1

HIV testing uptake and HIV positivity among presumptive tuberculosis patients in Mandalay, Myanmar, 2014-2017

PLOS ONE

Dear Khine Wut Yee Kyaw,

Thank you for submitting your manuscript to PLOS ONE. After careful consideration, we feel that it has merit but does not fully meet PLOS ONE’s publication criteria as it currently stands. Therefore, we invite you to submit a revised version of the manuscript that addresses the points raised during the review process. The manuscript requires significant editing to minimize the language errors.

We would appreciate receiving your revised manuscript by 25th May 2020. To enhance the reproducibility of your results, we recommend that if applicable you deposit your laboratory protocols in protocols.io, where a protocol can be assigned its own identifier (DOI) such that it can be cited independently in the future. For instructions see: http://journals.plos.org/plosone/s/submission-guidelines#loc-laboratory-protocols

We look forward to receiving your revised manuscript.

Kind regards,

Kwasi Torpey, MD PhD MPH

Academic Editor

PLOS ONE

Additional Editor Comments (if provided):

Thank you for the revisions. The comments have satisfactorily addressed. However there are still several language errors in the manuscript. I suggest the final manuscript is copyedited by a native speaker

---

## [Author Response · Author response to Decision Letter 1]

24 May 2020

Review comments

Reviewer: Joseph Mugisha Okello

Title for manuscript: HIV testing uptake and HIV positivity among presumptive tuberculosis patients in Mandalay, Myanmar, 2014-2017

General comment

1. This important manuscript covers an important subject on HIV testing uptake and HIV positivity among presumptive tuberculosis patients. As the authors rightly, say, HIV testing uptake is still low in patients with presumptive TB. Therefore this manuscript generates more literature on this important topic

Author’s response: Thank you for this encouraging comment.

2. The manuscript is well written, and coherent which makes it easy to read. My only concern is why the authors never followed up the 1770 patients that never tested for HIV. In the discussion, the authors talk about the reasons (from qualitative studies) why patients with TB do not test for HIV. However, since the study setting looks to be organised, it would have been better to follow up the 1770 patients (15%) that never tested for HIV to study the reasons why. 

Author’s response: We agreed that those patients with presumptive TB should be followed up or HIV testing should be offered on their subsequent visits. . However, our current program practice does not require presumptive TB patients to come for follow-up unless they have other health issues or they develop TB symptoms later on. So we have added a recommendation to conduct a further study to explore the reasons for not having HIV testing among presumptive TB patients. We have included this in the manuscript. [line number 346-347]

Specific comments

1. Data for this study was obtained from a TB program. Although the authors got ethical clearance from the National TB and HIV/AIDS programme and the ethics review committee, there is no mention in the manuscript on whether the authors received informed consent from the study participants/patients. 

 Author’s response: Thank you for this comment. As the study involved routinely collected program data, informed patient consent was waived by both ethical committees. We have included this in the manuscript. [line number 209-211]

2. It is important that the authors give some details on how patients co-infected with TB and HIV are managed or how and where they are referred.

Author’s response: Thank you. We have provided this information in the manuscript. [line number 150-156]

3. In addition, the authors should give details on how the program follows TB patients referred to the resident townships for continuation of anti-TB treatment. 

Author’s response: Thank you. We have revised this in the manuscript. [line number 154-156]

---

## [Editor Report · Decision Letter 2]

27 May 2020

HIV testing uptake and HIV positivity among presumptive tuberculosis patients in Mandalay, Myanmar, 2014-2017

PONE-D-19-33429R2

Dear Dr. Khine Wut Yee Kyaw

We are pleased to inform you that your manuscript has been judged scientifically suitable for publication and will be formally accepted for publication once it complies with all outstanding technical requirements.

With kind regards,

Kwasi Torpey, MD PhD MPH

Academic Editor

PLOS ONE
---

## [Editor Report · Acceptance letter]

5 Jun 2020

PONE-D-19-33429R2 

HIV testing uptake and HIV positivity among presumptive tuberculosis patients in Mandalay, Myanmar, 2014-2017 

Dear Dr. Kyaw:

I'm pleased to inform you that your manuscript has been deemed suitable for publication in PLOS ONE. Congratulations! Your manuscript is now with our production department. 

Kind regards, 

on behalf of

Professor Kwasi Torpey 

Academic Editor

PLOS ONE